# R$^3$ Prompting: Review, Rephrase and Resolve for Chain-of-Thought Reasoning in Large Language Models under Noisy Context

**Qingyuan Tian[1], Hanlun Zhu[1], Lei Wang[2], Yang Li[3], Yunshi Lan[1*]**

[1]East China Normal University, Shanghai, China
[2]Singapore Management University, Singapore
[3]Alibaba Group, Beijing, China
{qytian, hlzhu}@stu.ecnu.edu.cn, yslan@dase.ecnu.edu.cn
lei.wang.2019@phdcs.smu.edu.sg
ly200170@alibaba-inc.com

## Abstract

With the help of Chain-of-Thought (CoT) prompting, Large Language Models (LLMs) have achieved remarkable performance on various reasoning tasks. However, most of them have been evaluated under noise-free context and the dilemma for LLMs to produce inaccurate results under the noisy context has not been fully investigated. Existing studies utilize trigger sentences to encourage LLMs to concentrate on the relevant information but the trigger has limited effect on final answer prediction. Inspired by interactive CoT method, where intermediate reasoning steps are promoted by multiple rounds of interaction between users and LLMs, we propose a novel prompting method, namely R$^3$ prompting, for CoT reasoning under noisy context. Specifically, R$^3$ prompting interacts with LLMs to perform key sentence extraction, variable declaration and answer prediction, which corresponds to a thought process of reviewing, rephrasing and resolving. The responses generated at the last interaction will perform as hints to guide toward the responses of the next interaction. Our experiments show that R$^3$ prompting significantly outperforms existing CoT prompting methods on five reasoning tasks under noisy context. With `GPT-3.5-turbo`, we observe 3.7% accuracy improvement on average on the reasoning tasks under noisy context compared to the most competitive prompting baseline. More analyses and ablation studies show the robustness and generalization of R$^3$ prompting method in solving reasoning tasks in LLMs under noisy context.

## 1 Introduction

Recent advances in Large Language Models (LLMs) like GPT-3 (Brown et al., 2020), PaLM (Chowdhery et al., 2022), OPT (Zhang et al., 2022a), and LLaMa (Touvron et al., 2023) have revolutionized the landscape of natural language

processing. A series of studies have proved that multi-step reasoning tasks can be easily solved by LLMs via different prompting methods (Kojima et al., 2022; Wei et al., 2023; Wang et al., 2023; Zhang et al., 2022b; Zhou et al., 2022; Zheng et al., 2023; Shi et al., 2023). Chain-of-thought (CoT), as the representative prompting approach, narrows the gap between human intelligence and machine intelligence by applying the rationales to few-shot prompting (Zhou et al., 2022). It has achieved impressive results on arithmetical reasoning tasks (Kojima et al., 2022; Wei et al., 2023; Wang et al., 2023; Zhang et al., 2022b; Zhou et al., 2022; Zheng et al., 2023; Shi et al., 2023).

However, most CoT approaches are investigated under a noisy-free context, where all the information provided in the problem of interest is relevant to the final answer. While in real-world scenarios, it is frequent to encounter problems with irrelevant information. Shi et al. (2023) first defined the reasoning tasks in LLMs under noisy context. As shown in Figure 1, the noisy context in the problem of interest includes "*Helen baked* 12 *berry cookies yesterday*", which can easily distract LLMs. They further proposed Instructed-CoT prompt, which includes a trigger sentence instructing LLMs to ignore the irrelevant information given in a question ahead of one-turn response exemplar. Instructed-CoT prompt is intuitive to denoise context but it has limited effect on the final answer prediction.

Inspired by prior studies (Zhou et al., 2022), interactive CoT prompting approaches show that conducting multi-round of interaction between users and LLMs as hints is efficient in progressively guiding toward the correct answers for complex reasoning tasks (Zheng et al., 2023; Zhou et al., 2022). Least-to-Most (Zhou et al., 2022; Zheng et al., 2023) is a CoT prompting method that guides LLMs to decompose a complex problem into simple sub-questions. The prompt first decomposes the problem into subproblems, then the generated inter-

---

*Corresponding author

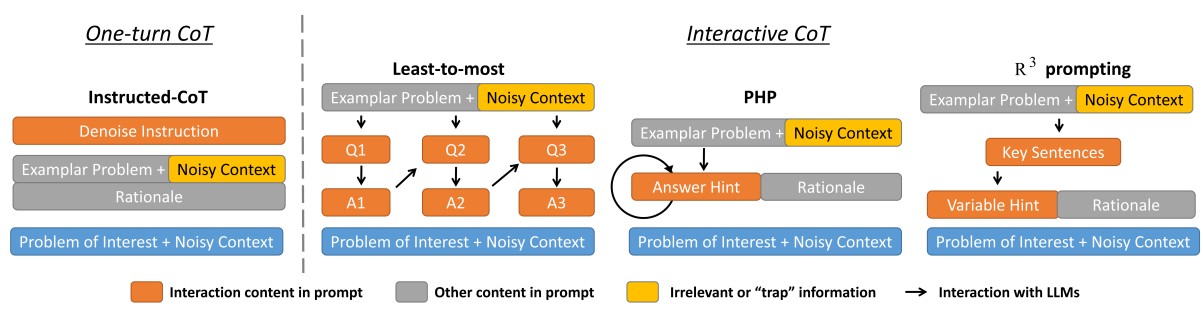

**[ Examplar Problem]** : Helen baked 19 chocolate cookies and 12 berry cookies yesterday, and she baked 231 raisin cookies and 237 chocolate cookiesthis morning. How many more chocolate cookies than raisincakes did Helen bake?

**[ Noisy Context ]** : Helen baked 12 berry cookies yesterday.

Figure 1: Comparison between $R^3$ prompting and existing CoT prompting baseline methods. The exemplar problems are multiple problems we used as exemplars for in-context learning. Rationales are reasoning chains in prompts. The problem of interest is the query problem.

mediate answer is appended to the next subproblem until the final answer is returned. PHP (Zheng et al., 2023) is another prompting method with interactive responses that combine the candidate answers and problems for re-evaluation purposes, which allows double-check to the answer prediction. Nevertheless, these interactive CoT prompting methods are still vulnerable to noisy context.

In this paper, we propose a new method named **$R^3$ prompting**, that includes sequentially interaction with LLMs to gradually approach final answers via a thought process of **Reviewing**, **Rephrasing** and **Resolving**. In the review stage, a review prompt is designed to extract key sentences that are essential conditions required for the final answer prediction. In the rephrase stage, we guide LLMs to reformulate the problem narratives to variables with the hint of extracted key sentences. In the resolve stage, LLMs predict the final answers taking account of the generated variables. All stages are implemented under few-shot setting with showing the responses of several exemplar problems. A comparison between the existing prompting methods is displayed in Figure 1.

We evaluate our prompting method on five different benchmarks containing misleading and irrelevant context, that are AddSub (Hosseini et al., 2014), SVAMP (Patel et al., 2021), MultiArith-IC, SingleEq-IC, and GSM-IC (Shi et al., 2023). The last three datasets are manually created by inserting noise to MultiArith (Roy and Roth, 2016), SingleEq (Koncel-Kedziorski et al., 2015) and GSM8K(Shi et al., 2023). We follow Shi et al. (2023) to create MultiArith-IC and SingleEq-IC by

randomly inserting irrelevant sentences into problem descriptions from MultiArith and SingleEq, respectively. Specifically, we follow their template-based method to create templates for noisy context and instantiate with role names in the problems. From the experimental results, we observe that interactive CoT has some advantages in solving noisy problems and our $R^3$ prompting method could outperform a series of CoT prompting methods for reasoning tasks in LLMs under noisy context, which leads to an average improvement of 3.7 absolute points on accuracy. More analyses shed lights on the robustness and generalization of $R^3$ prompting.

## 2 Related Work

**Chain-of-thought prompting.** CoT prompting (Wei et al., 2023) is a technique that guides LLMs to conduct multi-step reasoning which can be logically sequential without gradient accumulation. CoT has proved to be effective in solving reasoning tasks in few-shot scenarios. There are two major paradigms for CoT. One is to leverage a simple instruction sentence to elicit LLMs to do step-by-step thinking before answering a question (Kojima et al., 2022; Wang et al., 2023) or provide a few manual demonstrations with reasoning chains to show LLMs how to solve the exemplar problems (Wei et al., 2023; Zhang et al., 2022b). This paradigm is designed with one-turn response. Another paradigm of CoT has an interactive framework (Zheng et al., 2023; Zhou et al., 2022; Creswell et al., 2022; Yao et al., 2022), which enables automatic multiple rounds of interactions between users and LLMs by appending previously

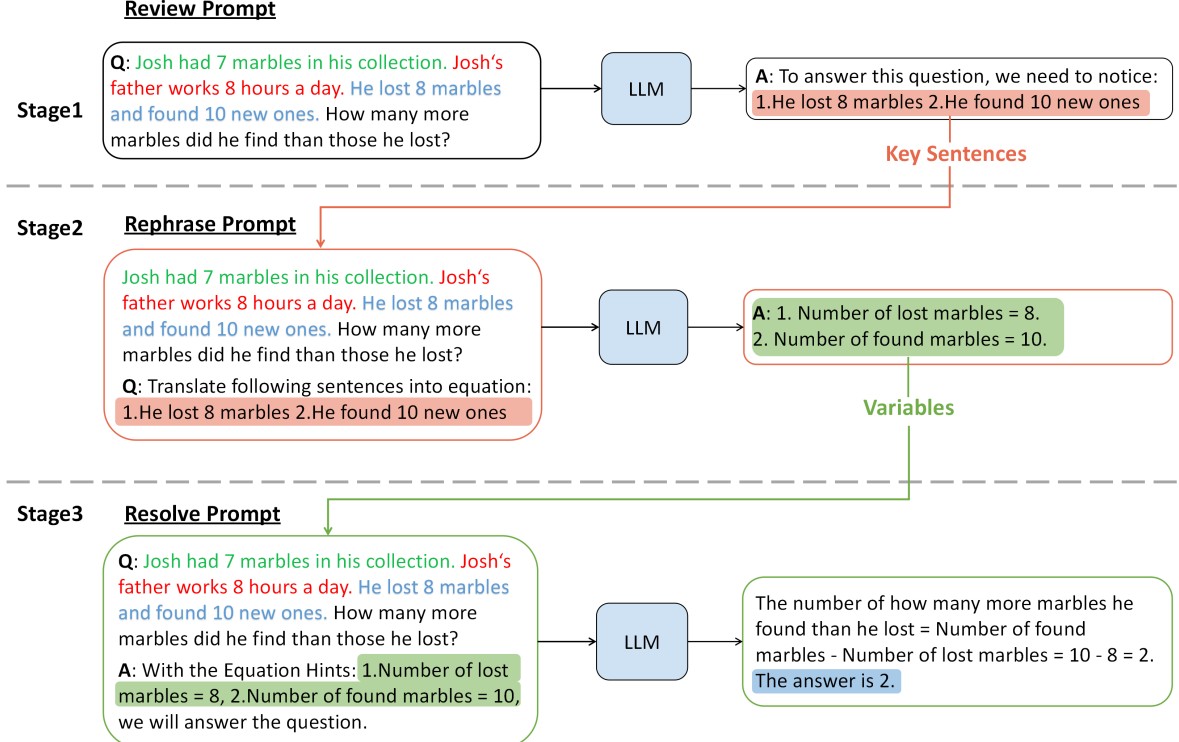

Figure 2: A running example of the inputs and outputs of R$^3$ prompting in LLMs at each prompting stage. Green: In-topic noisy context. Red: Off-topic noisy context. Blue: Key sentences.

generated. For example, Least-to-Most prompting (Zhou et al., 2022) interacts with LLMs via showing the decomposed sub-questions and corresponding solutions obtained from the last interaction, which enables complex reasoning. PHP prompting (Zheng et al., 2023) attaches the candidate answers generated from the last interaction to ask LLMs to double-check the answers. Our proposed method follows the interactive paradigm but is invested with a novel interactive process of thinking.

**Prompting with noisy ground truth.** A few studies have investigated the effect of prompts with noisy ground truth, including pairing exemplar problems with incorrect answers (Min et al., 2022; Kim et al., 2022) and adding irrelevant or misleading context into exemplars (Webson and Pavlick, 2021; Shi et al., 2023). In particular, Madaan and Yazdanbakhsh (2022) proved that the correctness of math equations in exemplars does not notably influence model's performance, whereas the noise included in the text could hamper the performance dramatically. Similar observation has been discussed in another work (Zhang et al., 2022b). To solve the reasoning task under noisy context, Shi et al. (2023) introduced an intuitive trigger sentence in prompts to decrease the influence from the noisy context to LLMs. Different from the simple instruction for denoising in their prompts, our work focuses on designing a interactive CoT prompt that can steadily solve problems without being misguided by noisy context.

## 3 R$^3$ Prompting

Researches on human problem-solving indicate that when solving a reasoning task under noisy contexts, the instinct of humanity is to conduct multiple rounds of thinking (Newell et al., 1959). In this paper, we propose that this process can be simulated in LLMs by reviewing, rephrasing, and resolving successively. Specifically, we design a prompting method with three-stage prompts: (1) **Review prompt**: instruct LLMs to read the question and extract the key sentences from the noisy contexts. (2) **Rephrase prompt**: with the hint of extracted key sentences, reformulate the narratives into abstractive representations with explicit variables. (3) **Resolve prompt**: with the hint of explicit variables, solve the question and predict the final answers. Figure 2 illustrates the proposed interaction and prompt design for each stage. Next, we will introduce the detailed design of prompts in

each stage.

## 3.1 Review Prompt

The goal of this stage is to review the input and distinguish the key sentences from the noisy context. Following the prior studies (Shi et al., 2023), we focus on two common categories of noisy information in contexts: 1) Deceptive in-topic sentences designed to complex the problem, as "Josh had 7 marbles in his collection" shown in Figure 2; 2) Off-topic sentences mixed to distract the solvers, as "Josh's father works 8 hours a day" shown in Figure 2. To prevent LLMs from being trapped by the noisy information, we provide exemplar problems and their expected responses before the problem of interest in prompts with the following design principles:

- The exemplar problem should include both categories of the above noisy information, which can elicit comprehensive reviewing ability of LLMs.

- The demonstrations should show the expected key sentences to be extracted by LLMs, which are the essential conditions needed to reason the answers.

To apply the first principle, we synthesize exemplar problems containing diverse noisy contexts. Specifically, we randomly sample several problems from the training data which includes in-topic noisy sentences. Then we write some templates for off-topic sentences which can be instantiated with different role names and we randomly insert them into the problems. For example, in Figure 2, we have exemplar problems with both types of noise "*Josh had 7 marbles in his collection*" and "*Josh's father works 8 hours a day*", which corresponds to in-topic and off-topic noisy sentences, respectively.

In order to apply the second principle, we organize the format of responses as:

```
To answer this question, we need to
notice: 1. [C₁] 2. [C₂] ...
```

where [$C_i$] denotes the $i$-th key sentence in the problem of interest that LLMs should focus on during reasoning. We arrange them via numerical indexes to guide LLMs to explicitly specify their order. Furthermore, to ensure the semantic integrity between the key sentences and the original problems, we consider directly extracting textual spans from the original problems as the key sentences.

As shown in Figure 2, "*He lost* 8 *marbles*" and "*He found* 10 *new ones*" are directly taken as the interactive response of the review prompt. These two sentences are also the essential conditions that we can use to solve the problem.

So, in the first round of interaction, given the exemplar problems, corresponding responses as well as the problem of interest, LLMs will learn to review a problem under noisy context by focusing on the key sentences.

## 3.2 Rephrase Prompt

In the second round of interaction, we ask LLMs to rephrase the narratives of the extracted sentences into abstractive representation, which explicates the variables and corresponding numeral values. To this end, we append the extracted key sentences behind the problem of interest to guide LLMs to translate these sentences into variables.

We organize the format of response as

```
1. [V₁] 2. [V₂] ...
```

where $V_i$ denotes the variables corresponding to $C_i$. In Figure 2, we have "*Number of lost marbles* = 8" and "*Number of found marbles* = 10" as output of the rephrase prompt. As opposed to existing zero-shot prompting methods (Wang et al., 2023) where a simple trigger sentence is included in the prompt as the instruction, rephrase prompt instructs LLMs by providing well-organized exemplar problems as demonstrations.

## 3.3 Resolve Prompt

The third round of interaction is to resolve the question and predict the answer based on the given variables. Similar with the previous prompt, we append the variables behind the original question and request LLMs to provide final answers and we organize the format of response as:

```
[Question of Interest] = [Equations
involving V₁, V₂, ...]. The answer is
[A].
```

where the equation shows the reasoning process and [A] denotes the final answer. We extract the arabic numeral as our prediction. In Figure 2, we have "2" as the predicted answer to the question. It is worth noting that even thought we conduct multiple rounds of interaction with LLMs, we include the same exemplar problems with different responses for each stage interaction in R³ prompting.

| Dataset | #Sample | Ave. in-topic | Ave. off-topic |
|---|---|---|---|
| GSM-IC | 1000 | 0.5 | 0.5 |
| MultiArith-IC | 600 | 0.5 | 0.5 |
| SingEq-IC | 508 | 0.48 | 0.52 |

Table 1: Details of constructed datasets. "Ave. in-topic" and "Ave. off-topic" denotes averge number of in-topic sentences and off-topic sentences, respectively.

## 4 Experimental Setup

### 4.1 Datasets

To demonstrate the efficacy of $R^3$ prompting method under noisy context, our evaluation is conducted on multiple datasets: (1) **AddSub** (Hosseini et al., 2014) and (2) **SVAMP** (Patel et al., 2021) are two challenges of arithmetic reasoning, where AddSub contains one-unknown arithmetic word problems for up-to-4 grad level students and SVAMP covers addition and subtraction arithmetic word problems. The problems of both these datasets involve noisy context that is designed to confuse problem solvers. These noisy sentences are often in-topic and highly related to the problem but play no role in solving the question. We further include (3) **GSM-IC**, which is a dataset introduced by Shi et al. (2023) and created for the investigation on the destructibility of LLMs. It consists of a subset of GSM8K dataset (Cobbe et al., 2021) and each problem is interrupted by manually inserting some irrelevant in-topic and off-topic sentences that are generated by templates. (4) **MultiArith-IC** and (5) **SingleEq-IC** are two datasets we constructed following the above procedure (Shi et al., 2023) to include more noisy context for each problem. The original problems are extracted from MultiArith (Roy and Roth, 2016) and SingleEq (Koncel-Kedziorski et al., 2015), respectively. Since most of them are noisy-free, we create templates for noisy context and instantiate with role names in the problems The statistics of those constructed dataset is shown in Table 1.

### 4.2 Baselines

We compare $R^3$ prompting with a set of few-shot CoT prompting baseline methods, which can be categorized into two groups based on the interaction with LLMs: (1) **Manual-CoT** (Zhang et al., 2022b) and **Auto-CoT** (Zhou et al., 2022) are two prompting methods that provide one-turn response in the prompt to show the reasoning process of exemplars to LLMs. **Instructed-CoT** (Shi et al., 2023) is also a one-turn response prompting method but it is designed for tasks with noisy context by including a trigger sentence. (2) **Least-to-Most** (Zhou et al., 2022) and **PHP** (Shi et al., 2023) are two prompting methods with interactive responses, which leverage the reasoning paths and candidate answers of LLMs to refine reasoning, respectively. We test their distractibility in the presence of noisy context as illustrated in Figure 1.

### 4.3 Implementation

We employ engines including `text-davinci-002`, `text-davinci-003` and `GPT-3.5-Turbo`[1] for experiments due to their strong CoT reasoning abilities (Brown et al., 2020; Kojima et al., 2022; Ouyang et al., 2022). The main results are obtained via `GPT-3.5-Turbo` API. For temperature, the temperature of all interactions of comparable prompting methods is set to 0 except for PHP prompting. Following setup in the original paper (Zheng et al., 2023), the temperature for the second interaction of PHP prompting is set to 0.7 and the others are set to 0. Meanwhile, following Wei et al. (2023), the total number of demonstrations $k$ is set to 8. We select the same problems as exemplars for all the comparable methods to make fair comparison.

## 5 Experimental Results

### 5.1 Main Results

We find that our proposed $R^3$ prompting enhances LLM's reasoning ability, especially when facing problems under noisy context.

**$R^3$ prompting performs well for CoT reasoning in LLMs under noisy context.** $R^3$ prompting consistently outperforms other few-shot prompting methods across all arithmetic reasoning datasets under noisy context. There is a minimum improvement of $2\%$ in accuracy on AddSub dataset and an overall average increase of $3.3\%$ in accuracy. Among them, it achieves the largest performance gain on SVAMP dataset, where it improves the second best result $83.6\%$ produced by Auto-CoT to $87.3\%$. Considering that most of the problems in SVAMP contain in-topic noisy sentences, we believe the improvement is from the accurate recognition to these sentences. Even though Instructed-CoT is designed with a denoising instruction, it does not provide explicit demonstrations to LLMs

---

[1] we apply GPT-3.5-Turbo-0301 from `https://platform.openai.com/docs/models/gpt-3-5`

| | Methods | SVAMP | MultiArith-IC | SingleEq-IC | AddSub | GSM-IC | Average |
|---|---|---|---|---|---|---|---|
| One-turn | Manual-CoT | 79.9 | 79.5 | 77.7 | 85.3 | 81.0 | 79.7 |
| | Auto-CoT | 83.6 | 79.7 | 77.6 | 88.0 | 81.5 | 82.1 |
| | Instructed-CoT | 81.3 | 80.1 | 78.2 | 87.3 | 82.0 | 81.8 |
| Interactive | Least-to-Most | 80.8 | 77.4 | 76.2 | 85.3 | 81.1 | 80.2 |
| | PHP | 83.1 | 80.0 | 79.0 | 85.3 | 85.1 | 82.5 |
| | R$^3$ Prompting (Ours) | **87.3** | **82.2** | **81.5** | **90.0** | **88.0** | **85.8** |

Table 2: Main result on five evaluated datasets. The best and second best results are boldfaced and underlined respectively.

such that LLMs struggle to learn what kind of information should be ignored. In contrast, R$^3$ prompting provides strong supervision on the final answer prediction by showing demonstrations (81.8% vs. 85.8%).

**The design of interactive prompts are important for denoising.** Comparing the overall performance of one-turn and interactive prompting methods, we notice there is a moderate advantage of interactive prompting methods for reasoning under noisy context. We believe this is because interactive CoT prompting methods provide more intensive guidance toward the answers for LLMs. Since the other interactive CoT prompting methods are not designed for the purpose of denoising, their improvement is limited. The three-stage prompts of R$^3$ prompting are elaborately designed to stimulate the human ability of thinking and this enables LLMs to learn a more comprehensive and efficient problem-solving skill under noisy context.

Moreover, to demonstrate the generalizability of our method, we also evaluate R$^3$ prompting on MultiArith, SingleEq and GSM8K. The results indicate that our method generally shows good results for reasoning in LLMs under noisy context. The complete results are provided in the Appendix A.1.

### 5.2 Performance with Diverse Settings

**Improvement of R$^3$ prompting is consistent over various engines.** We evaluate Manual-CoT, Instructed-CoT and R$^3$ prompting with various engines including `text-davinci-002` and `text-davinci-003` and display the results in Table 4. We observe that R$^3$ prompting generally outperforms Manual-CoT and Instructed-CoT under noisy context with various engines. Furthermore, the performance R$^3$ prompt can achieve better results with more powerful engine. On AddSub dataset, few-shot R$^3$ prompting method surpasses the SoTA results (Roy and Roth, 2016) which is obtained by training with full data (94.9% → 95.7%).

**Improvement of R$^3$ prompting is still significant with self-consistency.** We combine prompting methods with self-consistency strategy (Wang et al., 2022). We sample answers with numbers 5, 10, 20, 40 and draw Figure 3 to show the performance change with the increasing number of answers.

We discover that the improvement brought by R$^3$ prompting is not conflict with self-consistency. For example, on AddSub, when the sample number is 5, R$^3$ prompting (90.0%) outperforms Instructed CoT (85.3%) with an improvement of 4.7% absolute point. When the sample number increases to 40, this gain does not diminish, R$^3$ prompting still outperforms Instructed CoT by 4.1%. Similar trend occurs on SVAMP dataset. This indicates that the improvement brought by R$^3$ prompting and self-consistency is orthogonal. They play roles on the different aspects of prompting.

### 5.3 Ablation Study

To demonstrate the significance of all stages employed in R$^3$ prompting, we display the results of ablation study in Table 3 by testing different combinations of prompts.

**Stages of reviewing, rephrasing and resolving are all important.** Combination (1) is similar to traditional one-turn CoT method. Combination (2) and (3) yield average accuracy (84.1% and 82.0%) respectively, which are significantly higher than combination (1). This indicates that both review and rephrase prompts are important. Among them, review prompt provides a larger performance gain on average, this indicates that reviewing the problem and identifying the key sentences is important under the noisy context. Overall, including all the prompts achieves the best results, which demonstrates an improvement of 6.1 absolute points over combination (1).

**The effect of review and rephrase prompts vary over different datasets.** We notice that combina-

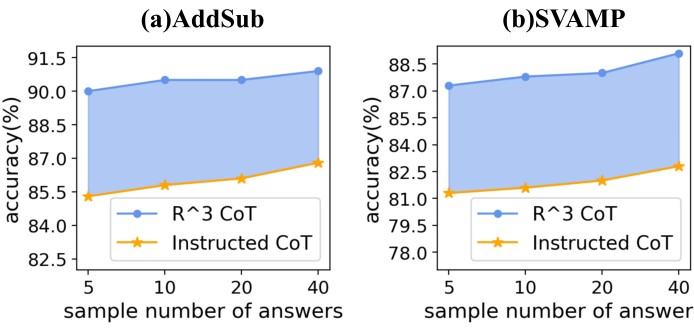

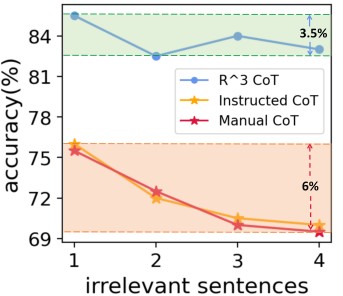

Figure 3: (a). The results of prompting methods after adding Self-Consistency (SC) on AddSub dataset. (b). The results of prompting methods after adding Self-Consistency (SC) on SVAMP dataset.

Figure 4: Accuracy change of various methods with the increasing number of irrelevant sentences on AddSub.

| | Stage | | | AddSub | MultiArith-IC | SingleEq-IC | SVAMP | GSM-IC | Average |
|---|---|---|---|---|---|---|---|---|---|
| | Review | Rephrase | Resolve | | | | | | |
| (1) | ✗ | ✗ | ✓ | 85.3 | 79.5 | 77.8 | 79.9 | 81.0 | 79.7 |
| (2) | ✓ | ✗ | ✓ | 88.6 | 81.0 | 78.9 | 84.4 | 83.0 | 84.1 |
| (3) | ✗ | ✓ | ✓ | 88.3 | 80.0 | 79.7 | 80.7 | 81.5 | 82.0 |
| (4) | ✓ | ✓ | ✓ | **90.0** | **82.2** | **81.5** | **87.3** | **88.0** | **85.8** |

Table 3: Ablation result of evaluation.

| Engine | Method | AddSub | GSM-IC |
|---|---|---|---|
| text-davinci-002 | Manual-CoT | 84.8 | 55.8 |
| | Instructed-CoT | 85.8 | 58.5 |
| | R$^3$ prompting | **87.0** | **59.6** |
| text-davinci-003 | Manual-CoT | 91.4 | 72.4 |
| | Instructed-CoT | 92.0 | 75.2 |
| | R$^3$ prompting | **95.7** | **78.1** |

Table 4: Performance of prompting methods on AddSub and GSM-IC datasets with different engines.

tion (3) shows more observable performance gain over combination (1) on AddSub and SingleEQ-IC datasets than the other datasets. In comparison, combination (2) shows more observable performance gain on AddSub and SVAMP than the other datasets. After investigating the datasets in detail, we find that SingleEQ-IC contains more questions requiring multi-step reasoning, GSM-IC contains more questions with noisy context and AddSub contains both. This highlights an intriguing finding that rephrasing is more crucial for complex reasoning and reviewing is more significant for denoising.

### 5.4 Effect on Different Noisy Context

To explore the effect of R$^3$ prompting on different numbers and types of noisy sentences, we first sample 200 problems from AddSub dataset, then introduce irrelevant sentences to each problem with numbers ranging from 1 to 4. This results

in datasets AddSub-1, AddSub-2, AddSub-3, and AddSub-4, where "-$n$" denotes the number of irrelevant sentences manually inserted in each problem. As shown in Figure 4, R$^3$ prompting maintains accuracy fluctuating around $84\%$ over all the datasets. In contrast, both Instructed-CoT and Manual-CoT experience a decrease in accuracy with the increasing amount of irrelevant sentences involved. It proves that R$^3$ prompting exhibits robust performance under noisy context while Instructed-CoT and Manual-CoT are vulnerable when facing a large amount of noisy information.

As we discussed before, this study focuses on two categories of noisy context, which are in-topic noise and off-topic noise. To investigate which type of noise R$^3$ prompting handles better, we collected 200 problems from the AddSub dataset and introduce different categories of irrelevant sentences. This form AddSub-in-topic and AddSub-off-topic datasets that contain a single type of noisy context. We discovered that R$^3$ achieves $92\%$ accuracy on AddSub-off-topic dataset, while it achieves an accuracy of $87.5\%$ on AddSub-in-topic dataset. This indicates that compared with off-topic noise, in-topic noise is more difficult to be resolved as it provides more deceptive information that can easily "trap" LLMs.

| Error Type | R³ |
|---|---|
| Calculation error | 50% |
| False negative in reviewing | 4% |
| False positive in reviewing | 36% |
| Rephrase errors | 10% |

Table 5: Error analysis of R³ prompting.

## 5.5 Error Analysis

To better understand how R³ prompting works under noisy context, we present an illustrative example in Figure 5 that is correctly predicted by R³ prompting but incorrectly predicted by other methods. As we can see, sentence "21 *dimes*" is a noisy sentence that is related to the problem narratives but irrelevant to the final answer prediction. Both Manual-CoT and Instructed-CoT produce wrong rationales that include the noisy sentence but this eventually results in a wrong predicted answer. For R³ prompting, after reviewing the problems, key sentences are extracted from the problem so that the rephrasing and resolving stages could escape from the disturbance of the noisy context and eventually output the correct answer. Interestingly, in the rephrase stage, the hint "*The number of dimes is irrelevant to this question.*" is considered as a weak requirement for a correct equation. Therefore, the LLMs can directly benefit and obtain a final output which is clear, concise, and free from unnecessary verbosity.

We further sample 50 error cases that R³ prompting have incorrectly predicted and analyzed them. As shown in Table 5, calculation error is the most common error accounting for around half of the total mistakes. This is caused by the fundamental mechanism of LLMs, where the probabilistic model is utilized to conduct calculation and may not exactly follow the mathematical calculation rules. This can be deemed as the upper boundary of solving reasoning tasks in LLMs. Regarding the errors caused by problem understanding and reasoning, compared with other baselines like Manual-CoT, R³ prompting method can already reduce them into a relatively low proportion (70% → 50%). Among them, errors caused by distinguishing wrong key sentences account for 40% of the total errors, which is significantly lower than Manual-CoT (66% → 40%). We also notice that false positive errors occur more frequently than false negative errors.

### Original Problem

**Q:** Mary had **21 dimes** and 38 pennies in her bank . Her dad borrowed 18 pennies from Mary . How many pennies does she have now ?

### Manual-CoT Output

Mary had 21 dimes and 38 pennies, so **she had a total of 21 + 38 = 59 coins.**
Her dad borrowed 18 pennies from her, so she now has 59 - 18 = 41 pennies.
Therefore, Mary has 41 pennies left.

### Instructed-CoT Output

Mary had 21 dimes and 38 pennies, so **she had a total of 21 + 38 = 59 coins.** Her dad borrowed 18 pennies from her, so she now has 59 - 18 = 41 pennies.
The answer is 41.

### R³ Prompting Output

We need to notice: 1."Mary had 38 pennies in her bank.", 2."Her dad borrowed 18 pennies from Mary.", 3."The number of dimes is irrelevant to this question.".

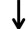

1. "Number of pennies Mary had = 38."
2. "Number of pennies borrowed by dad = 18."
3. "Number of pennies Mary has now = Number of pennies Mary had - Number of pennies borrowed by dad."

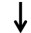

Number of pennies Mary has now = **Number of pennies Mary had - Number of pennies borrowed by dad** = 38 - 18 = 20. The answer is 20.

Figure 5: An illustrative problem from AddSub dataset, the result of which is incorrectly predicted by other prompting methods but correctly predicted by R³ prompting. Incorrect rationales and answers are displayed in red color. Correct rationales and answers are displayed in green color. Noisy context is displayed in blue color.

## 6 Conclusion

In this paper, we proposed R³ prompting to enable LLMs to solve problems under noisy context. It involves prompts for three different stages: Reviewing, Rephrasing, and Resolving. Evaluation on eight datasets across two types of noisy context suggests R³ prompting significantly outperforms the previous baselines. More analyses and ablation studies show that the performance of R³ prompting is stable and powerful. Overall, the advantages of R³ prompting are presented in the following aspects: 1) Achieving substantial performance improvement on arithmetic reasoning tasks under the interference of noisy context; 2) Exhibiting strong

robustness, maintaining stable performance even when facing progressively increasing amounts of noisy context.

## Limitations

Despite the improvement achieved by $R^3$ prompting, there are still aspects that can be improved. For instance, greedy decoding may not be the optimal strategy to produce the output answer as the best-first search may not result in the global best result. How to design an efficient strategy for sampling is a direction that many researchers are currently investigating (Wang et al., 2022; Weng et al., 2022; Li et al., 2022). In our method, the output in the review stage is currently obtained through greedy decoding. To achieve higher-quality review outcomes, future work could design a strategy, similar to Self-Consistency, to evaluate and sample the results from the review stage which are necessary and sufficient for problem-solving. Therefore, developing effective sampling strategies based on $R^3$ prompting would be an fascinating direction to explore.

## Ethics Statement

This is a study about Arithmetic Reasoning. It does not have any data privacy issues; we did not collect any personal information. This is a task that involved no risk as the participants are not exposed to any harmful material or asked to perform any risky tasks.

## Acknowledgments

The authors would like to thank the anonymous reviewers for their insightful comments. This work was supported by Natural Science Foundation of China (Project No. 62206097) and Shanghai Pujiang Talent Program (Project No. 22PJ1403000).

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

# A  Appendix

## A.1  Experiment Result on More Benchmarks

|  | Methods | AddSub | MultiArith | SingleEq | SVAMP | GSM8K | Avg. |
|---|---|---|---|---|---|---|---|
| One-turn | Manual-CoT | 85.3 | 97.2 | 92.9 | 79.9 | 78.9 | 86.8 |
|  | Auto-CoT | 88.0 | 98.2 | 92.3 | 83.6 | 79.9 | 88.4 |
|  | Instructed-CoT | 87.3 | 97.2 | 92.9 | 81.3 | 79.1 | 87.6 |
| Interactive | Least-to-Most | 85.3 | 89.8 | 85.0 | 80.8 | 79.3 | 84.0 |
|  | PHP | 85.3 | 98.0 | 92.9 | 83.1 | **84.5** | 88.8 |
|  | $R^3$ Prompting (Ours) | **90.0** | **98.2** | **93.1** | **87.3** | 80.1 | **89.7** |

Table 6: Accuracy of different methods on benchmarks with the engine of GPT-3.5-turbo

## A.2  $R^3$ Prompting Details

### A.2.1  $R^3$ Prompting I: REVIEW (7 Examples)

Q: Zachary did 46 push-ups and 58 crunches in gym class today. David did 38 more push-ups but 62 less crunches than Zachary. In the same gym class, there were 25 students in total. How many more crunches than push-ups did Zachary do?
A: We need to notice: 1."Zachary did 46 push-ups.", 2."Zachary did 58 push-ups.", 3."The number of crunches Zachary did is irrelevant to this question.", 4."The number of students is irrelevant to this question.".

Q: There are many different books in the 'crazy silly school' series. If you have read 13 of the books and are yet to read 8 books. The author of the series, Mr. Johnson, has written a total of 35 books. How many books are there in the 'crazy silly school' series?
A: We need to notice: 1."You have read 13 of the books.", 2."You are yet to read 8 books.", 3."The sum of read and unread books is equal to how many books are here.", 4."How many books did the author write is irrelevant to this question.".

Q: Every day Ryan spends 6 hours on learning english and 7 hours on learning chinese. During the weekend, Ryan also spends 3 hours on playing video games each day. If he learns for 5 days, how many hours does he spend on learning english and chinese in all?
A: We need to notice: 1."Every day Ryan spends 6 hours on learning english and 7 hours on learning chinese.", 2."He learns for 5 days.", 3."How long does Ryan spend on games is irrelevant to this question.".

Q: Matthew had 29 crackers and 30 cakes. The total number of cookies Matthew had was 15. If Matthew gave equal numbers of crackers and cakes to his 2 friends. How many cakes did each person eat?
A: We need to notice: 1."Matthew had 30 cakes.", 2."Matthew gave equal numbers of crackers and cakes to his 2 friends.", 3."The number of crackers and cookies is irrelevant to this question.".

Q: For Gwen's birthday she received 8 dollars from her mom. Her dad gave her 5 more dollars. The total number of guests at Gwen's birthday party was 12. If she spent 4 dollars. How much more money did she receive from her mom than she did from her dad?
A: We need to notice: 1."For Gwen's birthday she received 8 dollars from her mom.", 2."Her dad gave her 5 more dollars.", 3."The dollars she spent is irrelevant to this question.", 4."The number of guests is irrelevant to this question.".

Q: Jack received 5 emails and 6 letters in the morning. He then received 9 emails and 7 letters in the afternoon. The total number of text messages Jack received during the day was 24. How many letters did jack receive in the day?
A: We need to notice: 1."Jack received 6 letters in the morning.", 2."Jack received 7 letters in the afternoon.", 3."The number of emails and text messages Jack received is irrelevant to this question.".

Q: Paul had 108 books and 55 pens. After selling some books and pens in a garage sale he had 66 books and 59 pens left. Paul also sold 4 tables during the garage sale. How many books did he sell in the garage sale?
A: We need to notice: 1."Paul had 108 books.", 2."After selling some books in a garage sale he had 66 books left.", 3."The number of pens and tables is irrelevant to this question.".

## A.2.2   $R^3$ Prompting II: REPHRASE (4 Examples)

Jack received 5 emails and 6 letters in the morning. He then received 9 emails and 7 letters in the afternoon. The total number of text messages Jack received during the day was 24. How many letters did jack receive in the day?
Q: Translate following sentences into equation: 1.'Jack received 6 letters in the morning.' 2.'Jack received 7 letters in the afternoon.' 3.'The total number of letters jack received is what we need to find'
A: 1."Letters received in the morning = 6."
2."Letters received in the afternoon = 7."
3."Total number of letters = letters received in the morning + letters received in the afternoon = 6 + 7."

Mary is baking a cake. The recipe calls for 14 cups of flour and 6 cups of sugar. She already put in 5 cups of flour. How many more cups of flour than cups of sugar does she need to add now?
Q: Translate following sentences into equation: 1.'The recipe calls for 14 cups of flour and 6 cups of sugar.' 2.'She already put in 5 cups of flour.' 3.'The difference between the cups of flour and cups of sugar still needed is what we need to find.'
A: 1."Cups of flour needed = 14, Cups of sugar needed = 6."
2."Cups of flour already added = 5."
3."Difference = cups of flour still needed - cups of sugar still needed= cups of flour needed - cups of flour added - cups of sugar needed."

Every day Ryan spends 6 hours on learning english and 7 hours on learning chinese. During the weekend, Ryan also spends 3 hours on playing video games each day. If he learns for 5 days, how many hours does he spend on learning english and chinese in all?
Q: Translate following sentences into equation: 1.'Every day Ryan spends 6 hours on learning english and 7 hours on learning chinese.' 2.'He learns for 5 days.'
A: 1."Time for english in one day = 6 hours, Time for chinese in one day = 7 hours."
2."The total time of learning chinese and english = (Time for english in one day + Time for chinese in one day) * 5."

For Gwen's birthday she received 8 dollars from her mom. Her dad gave her 5 more dollars. If she spent 4 dollars. How much more money did she receive from her mom than she did from her dad?
Q: Translate following sentences into equation: 1.'Gwen's birthday she received 8 dollars from her mom.' 2.'Her dad gave her 5 more dollars.'
A: 1."dollars received from mom = 8."
2."dollars received from dad = 5."

### A.2.3 $R^3$ Prompting III: RESOLVE (8 Examples)

Q: Zachary did 46 push-ups and 58 crunches in gym class today. David did 38 more push-ups but 62 less crunches than Zachary. In the same gym class, there were 25 students in total. How many more crunches than push-ups did Zachary do?
A: With the Equation Hints: 'Push-ups number of Zachary = 46.', 'Crunches number of Zachary = 58.', 'More crunches number = Crunches number of Zachary - Push-ups number of Zachary = 58 - 46.', we will answer the question. Difference in crunches number and Zachary number = Crunches number of Zachary - Push-ups number of Zachary = 58 - 46. 58 - 46 = 12. The answer is 12.

Q: There are many different books in the 'crazy silly school' series. If you have read 13 of the books and are yet to read 8 books. The author of the series, Mr. Johnson, has written a total of 35 books. How many books are there in the 'crazy silly school' series?
A: With the Equation Hints: 'books (you have read) number = 13, books (yet to read) = 8.', 'books number in the series = books (you have read) number + books (yet to read) = 13 + 8.', we will answer the question. Books number in the series = books (you have read) number + books (yet to read) = 13 + 8. 13 + 8 = 21. The answer is 21.

Q: Every day Ryan spends 6 hours on learning english and 7 hours on learning chinese. During the weekend, Ryan also spends 3 hours on playing video games each day. If he learns for 5 days, how many hours does he spend on learning english and chinese in all?
A: With the Equation Hints: 'Time for english in one day = 6 hours, Time for chinese in one day = 7 hours.', 'The total time of learning chinese and english = (Time for english in one day + Time for chinese in one day)*5 = 13 * 5.', we will answer the question. The total time of learning chinese and english = (Time for english in one day + Time for chinese in one day) * 5 = 13 * 5. So, 13 * 5 = 65. The answer is 65.

Q: Mary is baking a cake. The recipe calls for 14 cups of flour and 6 cups of sugar. Mary's mother has already made 3 cakes before. She already put in 5 cups of flour. How many more cups of flour than cups of sugar does she need to add now?
A: With the Equation Hints: 'Cups of flour needed = 14, Cups of sugar needed = 6.', 'Cups of flour already added = 5.', 'Difference = cups of flour needed - cups of flour already added - cups of sugar needed = 14 - 5 - 6.', we will answer the question. There are 14 - 5 - 6 = 9 - 6 = 3 cups of flour than cups of sugar she need to add. The answer is 3.

Q: Shawn has five toys. For Christmas, he got two toys each from his mom and dad. How many toys does he have now?
A: With the Equation Hints: 'Shawn started with 5 toys.', 'Shawn got 2 * 2 = 4 toys.', we will answer the question. Shawn started with 5 toys. Then Shawn got 2 * 2 = 4 toys. 5 + 4 = 9. The answer is 9.

Q: Jack received 5 emails and 6 letters in the morning. He then received 9 emails and 7 letters in the afternoon. The total number of text messages Jack received during the day was 24. How many letters did jack receive in the day?
A: With the Equation Hints: 'Letters received in the morning = 6.', 'Letters received in the afternoon = 7.', 'Total number of letters = letters received in the morning + letters received in the afternoon = 6 + 7.', we will answer the question. Total number of letters = letters received in the morning + letters received in the afternoon = 6 + 7. So, 6 + 7 = 13. The answer is 13.

Q: For Gwen's birthday she received 8 dollars from her mom. Her dad gave her 5 more dollars. If she spent 4 dollars. How much more money did she receive from her mom than she did from her dad?
A: With the Equation Hints: 'Dollars received from mom = 8.', 'Dollars received from dad = 5.', we will answer the question. She received 8 - 5 = 3 dollar from her mom than from her dad. The answer is 3.

Q: Paul had 108 books and 55 pens. After selling some books and pens in a garage sale he had 66 books and 59 pens left. Paul also sold 4 tables during the garage sale. How many book s did he sell in the garage sale?
A: With the Equation Hints: 'Original number of books = 108.', 'Final number of books = 66.', 'Books sold = original number of books - final number of books = 108 - 66.', we will answer the question. The number of books he sold = original number of books - final number of books = 108 - 66 = 42. The answer is 42.

## A.3  Success Example Outputs by R$^3$ Prompting on Different Datasets

### A.3.1  SVAMP

**Question:**
Bobby ate 33 pieces of candy. Then he ate 4 more. He also ate 14 pieces of chocolate. How many pieces of candy and chocolate did Bobby eat altogether?

**Review stage output:**
We need to notice: 1."Bobby ate 33 pieces of candy.", 2."He ate 4 more pieces of candy.", 3."He also ate 14 pieces of chocolate.", 4."The total number of pieces of candy and chocolate is relevant to this question."

**Rephrase stage output:**
1."Pieces of candy eaten = 33." 2."Additional pieces of candy eaten = 4." 3."Pieces of chocolate eaten = 14." 4."Total number of pieces of candy and chocolate eaten = pieces of candy eaten + additional pieces of candy eaten + pieces of chocolate eaten."

**Resolve stage output:**
Total number of pieces of candy and chocolate eaten = pieces of candy eaten + additional pieces of candy eaten + pieces of chocolate eaten = 33 + 4 + 14 = 51. The answer is 51.

### A.3.2  MultiArith-IC

**Question:**
Wendy's grandmother sold 82 concert tickets. Wendy had 82 files on her computer. Wendy deleted 37 of them from computer and put the rest into folders with 5 files in each one. How many folders did Wendy end up with?

**Review stage output:**
We need to notice: 1."Wendy had 82 files on her computer.", 2."Wendy deleted 37 of them from her computer.", 3."The number of concert tickets sold by Wendy's grandmother is irrelevant to this question.", 4."The number of files in each folder is given as 5."

**Rephrase stage output:**
1."Number of files on Wendy's computer = 82." 2."Number of files deleted from Wendy's computer = 37." 3."Number of concert tickets sold by Wendy's grandmother = irrelevant." 4."Number of files in each folder = 5." 5."Number of folders Wendy ended up with = (Number of files on Wendy's computer - Number of files deleted from Wendy's computer) / Number of files in each folder."

**Resolve stage output:**
Number of folders Wendy ended up with = (Number of files on Wendy's computer - Number of files deleted from Wendy's computer) / Number of files in each folder = (82 - 37) / 5 = 9. The answer is 9.

### A.3.3  SingleEq-IC

**Question:**

Last year, David bought 10.0 books about science. This year, Vincent bought 10 books about animals, 1 book about outer space, and 3 books about trains. Each book cost $16. How much did Vincent spend on the books, this year?

**Review stage output:**

We need to notice: 1."Vincent bought 10 books about animals, 1 book about outer space, and 3 books about trains.", 2."Each book cost $16.", 3."How much David spent is irrelevant to this question."

**Rephrase stage output:**

1."Number of books Vincent bought = 10 + 1 + 3 = 14." 2."Cost of each book = $16." 3."Total cost of books Vincent bought = number of books Vincent bought * cost of each book = 14 * $16."

**Resolve stage output:**

Total cost of books Vincent bought = number of books Vincent bought * cost of each book = 14 * $16 = $224. The answer is $224.

### A.3.4  AddSub

**Question:**

Jason has 18 books and he has read 9 of them . Mary has 42 books . How many books do they have together ?

**Review stage output:**

We need to notice: 1."Jason has 18 books and he has read 9 of them.", 2."Mary has 42 books.", 3."The number of books they have read together is irrelevant to this question."

**Rephrase stage output:**

1."Number of books Jason has = 18, Number of books Jason has read = 9." 2."Number of books Mary has = 42." 3."Total number of books they have together = Number of books Jason has + Number of books Mary has = 18 + 42."

**Resolve stage output:**

Total number of books they have together = Number of books Jason has + Number of books Mary has = 18 + 42. So, 18 + 42 = 60. The answer is 60.

### A.3.5  GSMIC

**Question:**

Jackie spends 8 hours working, 3 hours of exercise and spends 8 hours of sleep per day. Jackie's sister is 3 years old. How much free time does Jackie have?

**Review stage output:**

We need to notice: 1."Jackie spends 8 hours working, 3 hours of exercise and spends 8 hours of sleep per day.", 2."The age of Jackie's sister is irrelevant to this question.".

**Rephrase stage output:**

1."Total hours in a day = 24." "Hours spent working = 8, Hours spent exercising = 3, Hours spent sleeping = 8." "Total hours spent = Hours spent working + Hours spent exercising + Hours spent sleeping = 8 + 3 +

8." "Free time = Total hours in a day - Total hours spent = 24 - (8 + 3 + 8)." 2. "The age of Jackie's sister is irrelevant to this question."

**Resolve stage output:**
Total hours spent = Hours spent working + Hours spent exercising + Hours spent sleeping = 8 + 3 + 8 = 19. Free time = Total hours in a day - Total hours spent = 24 - (8 + 3 + 8) = 5. The answer is 5.