# OpenReview forum: "R$^3$ Prompting: Review, Rephrase and Resolve for Chain-of-Thought Reasoning in Large Language Models under Noisy Context"
_EMNLP/2023/Conference — EMNLP 2023 Findings_

### Official Review · Reviewer_FVdJ · 2023-08-01

**Soundness:** 3

**Excitement:**

3: Ambivalent: It has merits (e.g., it reports state-of-the-art results, the idea is nice), but there are key weaknesses (e.g., it describes incremental work), and it can significantly benefit from another round of revision. However, I won't object to accepting it if my co-reviewers champion it.

**Paper Topic And Main Contributions:**

This research paper addresses the problem of reasoning performance in Large Language Models (LLMs) in the presence of noisy or irrelevant information. Noise can significantly degrade the performance of an LLM, leading to suboptimal responses. Current LLMs often fail to effectively extract essential information from the context, especially when the context is polluted with irrelevant or misleading information.
To tackle this problem, the authors propose a novel method known as R3 (Review, Rephrase, Resolve) prompting. This methodology enhances the reasoning ability of LLMs, especially in noisy contexts. The authors leverage the Chain-of-Thought (CoT) reasoning approach to implement the R3 prompting strategy.
The main contributions of this paper are as follows:
1)Proposes a novel method (R3 prompting) for improving the reasoning performance of LLMs in noisy contexts.
2)Demonstrates the application of the R3 prompting strategy across various problem types through several examples.
3)Shows the effectiveness of the R3 prompting strategy by conducting experiments and comparing the results with other methods. The authors provide empirical evidence that R3 prompting significantly outperforms other methods on various benchmarks.
4)Broadens the understanding of how to improve the performance of LLMs in the presence of noise, providing a new perspective on LLM reasoning.

**Reasons To Accept:**

1)The paper offers an innovative, multi-stage design with R3 Prompting, which emulates human reasoning processes.
2)The authors present evidence of their method's superiority over certain Chain-of-Thought (CoT) methods, such as Manual-CoT and Instructed-CoT.
3)The authors candidly acknowledge the limitations of their methodology, which adds a degree of credibility to their research.

**Reasons To Reject:**

1)The paper's evaluation scope is quite narrow. A comprehensive understanding of the method's versatility could be gained by testing it across a wider range of tasks.
2)The lack of in-depth ablation studies on each stage of R3 prompting prevents a clear understanding of their individual contributions to the overall performance.
3)The paper lacks complex examples and a detailed error analysis, which could offer a more nuanced understanding of the method's capabilities and limitations.
4)There is a conspicuous absence of comparison with related works, particularly those involving interactive prompting.
5)The document could provide more information on how the R3 prompting strategy might be adapted or modified for different types of problems or different models.
6)The authors could also explore more about how this strategy could be incorporated into existing LLM training strategies to further enhance performance.

**Reproducibility:**

5: Could easily reproduce the results.

**Reviewer Confidence:**

3: Pretty sure, but there's a chance I missed something. Although I have a good feel for this area in general, I did not carefully check the paper's details, e.g., the math, experimental design, or novelty.

---

> ### Author Rebuttal · Authors · 2023-08-29
>
> We thank Reviewer FVdJ for positive comments and precious feedback on our work. We response specific comments below.
>
> **Q1**: The paper's evaluation scope is quite narrow. A comprehensive understanding of the method's versatility could be gained by testing it across a wider range of tasks.
>
> **A1**: We thank the reviewer for raising this point. Because our research falls within the CoT category, we followed the established standard settings of this field in experiment section, which is the same setting as [1], [2], [3], [4], [5] and so on. As shown in Table A in the paper's appendix, we initially conducted experiments on five datasets (AddSub, MultiArith, SingleEq, SVAMP, GSM8K) to demonstrate the versatility of our method. Furthermore, as we aimed to showcase the excellent noise resilience of our method in the face of irrelevant context, we followed the approach outlined in [1] to construct datasets (MultiArith-IC, SingleEq-IC, GSMIC). The experimental results for these datasets can be found in Table 2 in Section 5.2 of our paper.
>
> In summary, the paper's evaluation scope not only follows the experimental setups of related works but also introduces newly constructed tasks involving irrelevant context. And thanks for your suggestions, in our revised version, we will try more tasks like symbolic reasoning, commonsense reasoning and so on.
>
> [1]Shi, Freda, et al. "Large language models can be easily distracted by irrelevant context." International Conference on Machine Learning. PMLR, 2023.
>
> [2]Wei, Jason, et al. "Chain-of-thought prompting elicits reasoning in large language models." Advances in Neural Information Processing Systems 35 (2022): 24824-24837.
>
> [3]Zhang, Zhuosheng, et al. "Automatic chain of thought prompting in large language models." arXiv preprint arXiv:2210.03493 (2022).
>
> [4]Zhou, Denny, et al. "Least-to-most prompting enables complex reasoning in large language models." arXiv preprint arXiv:2205.10625 (2022).
>
> [5]Zheng, Chuanyang, et al. "Progressive-hint prompting improves reasoning in large language models." arXiv preprint arXiv:2304.09797 (2023).
>
> **Q2**: The lack of in-depth ablation studies on each stage of R3 prompting prevents a clear understanding of their individual contributions to the overall performance.
>
> **A2**: We thank the reviewer for raising this point. From Table 4 in Section 5.3, it's evident that each stage of R3 prompting plays an indispensable role. Moreover, the next stage depends to some extent on the previous stage, as average accuracy on combination 1 (79.7%) $\lt$ combination 2 (84.1%) $\lt$ combination 4 (85.8%). As for the individual roles of each stage, the review stage is the most crucial for filtering out irrelevant information, as evidenced by accuracy of combination 2 on the four datasets surpassing combination 3 (e.g. On AddSub 88.6% $\lt$ 88.3%). On the other hand, the rephrase stage is more beneficial for complex reasoning, as it essentially pre-processes information before problem-solving. Because these two points are intuitive and easily explainable, we did not provide overly detailed explanations in the paper. Overall, by combining these three stages, we achieved the best performance.
>
> **Q3**: The paper lacks complex examples and a detailed error analysis, which could offer a more nuanced understanding of the method's capabilities and limitations.
>
> **A3**: We thank the reviewer for raising this point. Regarding our method's capabilities and limitations, we provided detailed explanations of these aspects in Section 5.5 and the Limitations section of the paper. The primary concerns relate to the efficiency and accuracy of the review stage, with more complex examples or detailed error analyses largely associated with the first stage. However, we believe that employing sampling strategies akin to Self-consistency might significantly enhance the efficiency of this method.
>
> We will consider including more detailed error examples in the appendix in future versions.
>
> **Q4**: There is a conspicuous absence of comparison with related works, particularly those involving interactive prompting.
>
> **A4**: We thank the reviewer for raising this point. The main distinctions between our approach and other interactive prompting methods are as follows:
> 1. Different prompt design. Our approach simulates a general human thinking process. As shown in Figure 1, We employed a workflow-based approach to design this human-like prompting method. In contrast, other interactive prompting methods often rely on a multi-turn dialogue approach. For instance, PHP requires multiple iterations to refine the answer until a confidence threshold is met. Similarly, Least-to-most involves a multi-turn question-and-answer process until a final answer is obtained. Therefore, from a design perspective, the differences between these methods are quite distinct.
> 2. Different performance. As shown in experiments in Section 5.1 and 5.4, our model is mainly designed for reasoning tasks involving noisy context. Experiment results show the superiority of our method and its robustness as interference increases. Additionally, as mentioned, our method also outperforms PHP and Least-to-most in typical reasoning tasks.
>
> **Q5**: The document could provide more information on how the R3 prompting strategy might be adapted or modified for different types of problems or different models.
>
> **A5**: We thank the reviewer for raising this point. Firstly, our method can be easily applied to other large language models, as demonstrated in Table 4 of the paper. Secondly, our method is a human-like, universal approach, making it highly adaptable to other tasks. It might require some modifications to the prompt descriptions for different stages when applied to other tasks to meet specific requirements, but the functionality of each stage remains unchanged.
>
> **Q6**: The authors could also explore more about how this strategy could be incorporated into existing LLM training strategies to further enhance performance.
>
> **A6**: We thank the reviewer for raising this point. However, for a paper that utilizes prompting learning as its fundamental method, we believe that LLM training strategies do not fall within the scope of discussion. Nevertheless, we are excited to see future work incorporating this strategy into the training of LLMs.

---

### Official Review · Reviewer_56UP · 2023-08-01

**Soundness:** 4

**Excitement:**

2: Mediocre: This paper makes marginal contributions (vs non-contemporaneous work), so I would rather not see it in the conference.

**Paper Topic And Main Contributions:**

The paper investigates the problem of CoT prompting under noise context and propose an interactive prompting method termed $\mathrm{R^3}$ prompting. This method contains three stages: reviewing, rephrasing, and resolving. These processes collaboratively work to extract key sentences, reformulate the problem narratives to variables, and subsequently predict the final answers. Empirical results from five distinct datasets demonstrate the effectiveness of the proposed method.

**Questions For The Authors:**

Issues with reproducing the example in the error analysis. In the presented error analysis example in Figure 5, reviewing process in $\mathrm{R^3}$ prompting fails to recognize "She missed 12 games this year" as irrelevant, but its corresponding variable doesn't appear in the rephrase stage. As such, I reproduce rephrase process based on the rephrase prompt provided in the Appendix of the paper, LLM models fails to generate outputs as shown in the error analysis. The temperature is set as 0.

---
Reproduced outputs

gpt-3.5-turbo-0301
>A: 1."Number of soccer games attended this year = 11."
2."Number of soccer games missed this year = 12."
3."Number of soccer games attended last year = 13."
4."Number of soccer games planned to attend next year = 15."
Total number of soccer games attended = Number of soccer games attended this year + Number of soccer games attended last year + Number of soccer games planned to attend next year = 11 + 13 + 15."

text-davinci-003
>A: 1."Soccer games attended this year = 11."
2."Soccer games missed this year = 12."
3."Soccer games attended last year = 13."
4."Soccer games planned to attend next year = 15."
5."Total soccer games attended = soccer games attended this year + soccer games attended last year + soccer games planned to attend next year - soccer games missed this year."

text-davinci-002
>A: 1."Games this year = 11.", 2."Games missed this year = 12.", 3."Games last year = 13.", 4."Games next year = 15."
5."Total games = games this year + games last year + games next year - games missed this year."

**Reasons To Accept:**

- $\mathrm{R^3}$ prompting is effective, as evidenced by marked improvements when compared to baseline methods.
- The paper is easy to follow and the experimental results are thoroughly analyzed.

**Reasons To Reject:**

- The paper describes incremental work and the contribution is limited. The core contribution seems to be the fusion of interactive CoT prompting with CoT reasoning in the presence of noisy contexts. Given that both of these domains are already established and naturally aligned, the integration does not come across as particularly novel. In addition, the paper does not seem to tackle new problems or deliver significant new insights into the CoT reasoning under noisy context. For a submission to EMNLP, one might expect a deeper exploration.
- In [1], which serves as the primary reference for this paper, an interactive CoT prompting method (least-to-most prompting) is adopted within the context of CoT reasoning under noise. This method, designated as Instructed LTM, achieves best performance in [1]. However, in this paper, the $\mathrm{R^3}$ prompting is benchmarked only against a sub-optimal model in [1], the Instructed CoT, without any comparison to the Instructed LTM.

---
[1] Shi, Freda, et al. "Large language models can be easily distracted by irrelevant context." International Conference on Machine Learning. PMLR, 2023.

**Reproducibility:**

4: Could mostly reproduce the results, but there may be some variation because of sample variance or minor variations in their interpretation of the protocol or method.

**Reviewer Confidence:**

3: Pretty sure, but there's a chance I missed something. Although I have a good feel for this area in general, I did not carefully check the paper's details, e.g., the math, experimental design, or novelty.

---

> ### Author Rebuttal · Authors · 2023-08-29
>
> We thank Reviewer 56UP for positive comments and precious feedback on our work. We response specific comments below.
>
> **Q1**: The paper describes incremental work and the contribution is limited. The core contribution seems to be the fusion of interactive CoT prompting with CoT reasoning in the presence of noisy contexts. Given that both of these domains are already established and naturally aligned, the integration does not come across as particularly novel.
>
> **A1**: We thank the reviewer for raising this point. We would like to explain that the novelty of our work lies in following aspects.
> 1. The core contribution is the design of an interactive CoT method that aligns with human reasoning thinking patterns. The essence of nearly every interactive CoT method is to integrate interactive CoT prompts with CoT reasoning, such as PHP and Least-to-most. But why can't other interactive CoT methods effectively distinguish noisy contexts, while our method is highly effective in both noisy and noiseless contexts? We believe the core reason lies in the fact that our interactive mode aligns more with human thinking patterns. Human thinking is a deliberate and thoughtful decision-making system, as system 2 in [1], which allows it to better handle more complex reasoning problems.
> 2. Research of irrelevant context in reasoning is a novel and highly practical endeavor. As we all know, early CoT work has achieved significant success in simple reasoning tasks. However, as shown in the last research [2], it was discovered that the reasoning ability of LLMs can be easily disrupted by irrelevant information, leading to a significant drop in performance. As demonstrated in Table A1 in the paper's appendix, the existence of such irrelevant information has led to a 13% decrease in the performance of studies like PHP (98% on multiArith → 85.3% on Addsub). This highlights the necessity of enhancing LLMs ability for noisy context reasoning. However, aside from [2], there is currently limited research exploring this issue further.
>
> Notably, Reviewer 1 acknowledges the novelty of our idea, and Reviewer 3 agrees that our method's superiority for unlocking LLM’s in-context learning ability for reasoning tasks.
>
> [1] Yao, Shunyu, et al. "Tree of thoughts: Deliberate problem solving with large language models." arXiv preprint arXiv:2305.10601 (2023).
>
> [2] Shi, Freda, et al. "Large language models can be easily distracted by irrelevant context." International Conference on Machine Learning. PMLR, 2023.
>
> **Q2**: The paper does not seem to tackle new problems or deliver significant new insights into the CoT reasoning under noisy context.
>
> **A2**: We do believe there is a misunderstanding. This paper focuses on a new defined problem that is improving LLMs under noisy context and ends with some useful conclusions. Specifically, our method can generalize to solve reasoning problems containing noise. As we mentioned in last question, reasoning with irrelevant information represents a novel and more challenging category of tasks. As shown in Table 2 and Table A1, R3 prompting is an efficient and versatile method that outperforms other methods on datasets with and without irrelevant information. Additionally, Figure 4 demonstrates that as irrelevant information interference increases, this method exhibits strong robustness.
>
> Furthermore, our research has yielded some insightful conclusions. As described in Section 5.5, LLMs often struggle to identify these traps during the review stage, leading to errors. Therefore, how to encourage LLMs to recognize traps becomes crucial. This applies not only to reasoning tasks but is equally important in tasks like misinformation detection, question answering, translation, and more.
>
> **Q3**: Didn't use Instructed LTM as a benchmark.
>
> **A3**: We thank the reviewer for raising this point. Actually, we have conducted experiments with Instructed LTM, as shown in the following table. The results indicate that this approach inherits the drawbacks of LTM in simpler reasoning tasks, where deliberate decomposition of problems can lead to a significant performance decline. An intuitive observation is that while LTM achieves perfect performance on GSM8K, its performance drops on AddSub. Instructed LTM follows a similar pattern. Thanks for your suggestions, in our revised version, we will add Instructed LTM into the second section in Table 2 to give a more comprehensive evaluation. But it is worth noting our method still surpasses Instructed LTM on those five datasets.
>
> |                     | SVAMP | MultiArith-IC | SingleEq-IC | AddSub | GSM-IC |
> | :-----------------: | :---: | :-----------: | :---------: | :----: | :----: |
> |   Instructed LTM    | 81.1  |     78.3      |    77.2     |  85.6  |  84.5  |
> | R3 Prompting (Ours) | 87.3  |     82.2      |    81.5     |  90.0  |  88.0  |
>
> **Q4**: Question about the output.
>
> **A4**: We are so sorry if we make any confusion in writing part. Actually, Figure 5 is not a complete output demonstration. And your output is absolutely right, but in the process of figure making, due to space constraints and other reasons, we omitted the phrases "Number of soccer games missed this year = 12." and "Total number...13 + 15."
>
> The omission of the first sentence was because it constituted a transcription of irrelevant information. Since the transcription results of irrelevant information were not used in the resolve stage, and due to space limitations, we decided to remove it.
>
> The reason for omitting the second sentence is: After obtaining the initial output from the rephrase stage, we use regular expressions to remove extra sentences and retain only the sentences within double quotation marks. Therefore, to avoid unnecessary ambiguity in the figure, we decided to remove it.
>
> After you raised this question, we realized that we chose an inappropriate example for illustration, especially since the first stage fails to recognize noisy context. It also caused confusion for the readers in the figure. We sincerely apologize for that. Here is a better example in which each stage has perfectly achieved its intended goals:
>
> **Target question**：Jason has 18 books and he has read 9 of them . Mary has 42 books . How many books do they have together ?
>
> **Review output**：We need to notice: 1. "Jason has 18 books.", 2. "Mary has 42 books.", 3. "The number of books they have read is irrelevant to this question."
>
> **Rephrase output**：1."Number of books Jason has = 18." 2."Number of books Mary has = 42." 3."Total number of books they have together =Number of books Jason has + Number of books Mary has."
>
> **Resolve output**：Total number of books they have together = Number of books Jason has + Number of books Mary has = 18 + 42 = 60. The answer is 60.
>
> In the example above, the first stage correctly identifies "The number of read book" as irrelevant information. Secondly, during the rephrase stage, the LLMs directly outputs the reason for irrelevance, which is the correct problem-solving equation. Since this output is used as a hint-prompt input in the third stage, the LLMs can directly benefit and obtain the correct result in the resolve stage. Therefore, the last resolve process is clear, concise, and free from unnecessary verbosity. We will replace Figure 5 with this example in revised versions.

---

### Official Review · Reviewer_k3pf · 2023-08-05

**Soundness:** 4

**Excitement:**

3: Ambivalent: It has merits (e.g., it reports state-of-the-art results, the idea is nice), but there are key weaknesses (e.g., it describes incremental work), and it can significantly benefit from another round of revision. However, I won't object to accepting it if my co-reviewers champion it.

**Paper Topic And Main Contributions:**

LLMs are easily distracted by irrelevant context, leading to lower performance. This paper presents a prompting method for mitigating this issue. The method involves: (1) extracting relevant context; (2) rephrasing the context into hints; (3) getting the answers based on both original problems and hints. Experiments show the effectiveness of the proposed method.

**Reasons To Accept:**

The proposed method is straightforward and intuitive.

**Reasons To Reject:**

The contributions of this paper are incremental. This paper targets a narrow area of irrelevant context in reasoning. The proposed approach of incorporating extracting and rephrasing steps is very straightforward and does not offer substantial improvements over existing methods.

**Reproducibility:**

4: Could mostly reproduce the results, but there may be some variation because of sample variance or minor variations in their interpretation of the protocol or method.

**Reviewer Confidence:**

3: Pretty sure, but there's a chance I missed something. Although I have a good feel for this area in general, I did not carefully check the paper's details, e.g., the math, experimental design, or novelty.

---

> ### Author Rebuttal · Authors · 2023-08-29
>
> We thank Reviewer k3pf for positive comments and precious feedback on our work. We response specific comments below.
>
> Q: The contributions of this paper are incremental. This paper targets a narrow area of irrelevant context in reasoning. The proposed approach of incorporating extracting and rephrasing steps is very straightforward and does not offer substantial improvements over existing methods.
>
> A: We thank the reviewer for raising this point. We would like to explain that the main contributions of our work lie in following aspects.
> 1. Research of irrelevant context in reasoning is a novel and highly practical endeavor. As we all know, early CoT work has achieved significant success in simple reasoning tasks. However, as shown in the latest research [1], it was discovered that the reasoning ability of LLMs can be easily disrupted by irrelevant information, leading to a significant drop in performance. As demonstrated in Table A1 in the paper's appendix, the existence of such irrelevant information has led to a 13% decrease in the performance of studies like PHP (98% on MultiArith → 85.3% on AddSub). This highlights the necessity of enhancing LLMs ability for noisy context reasoning. However, aside from [1], there is currently limited research exploring this issue further.
> Therefore, our research aims to address the shortcomings of current CoT research, rather than simply transplanting existing approaches into a completely new domain.
> 2. Our method may appear simple and straightforward, but it has been carefully designed. By simulating the human reasoning-solving patterns, we have developed this concise yet efficient method. As shown in Figure 1, our method follows an interactive CoT manner but with different prompting strategies. Our goal is to impart general reasoning capabilities to LLMs, not just eliminating interference. As demonstrated in Appendix Table A1, our method outperforms other baseline methods, even in reasoning problems without interference, showcasing its universality.
> 3. Our research has shown general improvement on a series of reasoning tasks and yielded insightful conclusions that could inspire future work. As described in Section 5.5, reasoning problems with trap sentences are often more challenging. LLMs frequently struggle to identify these traps during the review stage, leading to errors. Therefore, we suggest future research could explore better strategies for handling trap information.
>
> [1] Shi, Freda, et al. "Large language models can be easily distracted by irrelevant context." International Conference on Machine Learning. PMLR, 2023.

---

### Meta-Review · Area_Chair_566o · 2023-09-27

**Recommendation:** 3

**Metareview:**

This paper presents a three-stage interactive prompting approach to ensure that LLMs perform well in reasoning tasks in noisy contexts. The proposed method is simple and effective as shown in the experiments on several datasets. The contributions are somewhat slim though, the paper can benefit from having the evaluation across more tasks and having more error analyses.

---

### Decision · Program_Chairs · 2023-10-07

**Decision:**

Accept-Findings

**Comment:**

This paper presents a three-stage interactive prompting approach to ensure that LLMs perform well in reasoning tasks in noisy contexts. The proposed method is simple and effective as shown in the experiments on several datasets. The contributions are somewhat slim though, the paper can benefit from having the evaluation across more tasks and having more error analyses.